# Hormonal Crosstalk in Melasma: Unraveling the Dual Roles of Estrogen and Progesterone in Melanogenesis

**DOI:** 10.3390/ijms262210856

**Published:** 2025-11-08

**Authors:** Jian Zhang, Tao Wang, Zhixian Li, Chuntang Qin, Jinjin Dai, Yihan Zhao, Shiguo Wu, Zhuangzhuang Jia

**Affiliations:** 1School of Basic Medical Sciences, Yunnan University of Chinese Medicine, Kunming 650500, China; 18088098105@163.com (J.Z.); pshbolo@163.com (Z.L.); qinchuntang@163.com (C.Q.); daiijinn@163.com (J.D.); 13769005886@163.com (Y.Z.); 2Yunnan Key Laboratory of Integrated Traditional Chinese and Western Medicine for Chronic Disease in Prevention and Treatment, Kunming 650500, China; 3Key Laboratory of Microcosmic Syndrome Differentiation, Education Department of Yunnan, Kunming 650500, China; 4School of Nursing, Yunnan University of Chinese Medicine, Kunming 650500, China; y2275861600@163.com(T.W.)

**Keywords:** melasma, estrogen, progesterone

## Abstract

Melasma is a commonly acquired hyperpigmentation disorder characterized by symmetrical facial macules, with a notably higher prevalence in women and individuals with darker skin tones. Its etiology involves a multifactorial interplay of genetic predisposition, ultraviolet radiation exposure, and hormonal factors. This review consolidates current evidence highlighting the instrumental roles of estrogen and progesterone in modulating melanogenesis. These hormones regulate melanocyte activity via genomic and non-genomic signaling pathways, impacting key enzymes and transcription factors critical to pigment synthesis. Furthermore, crosstalk between hormonal signaling cascades exacerbates hyperpigmentation, contributing to the development of melasma. Therapeutically, targeting endocrine pathways is a promising yet investigational approach, as long-term clinical data remain scarce. Interventions such as receptor modulators and metabolic inhibitors may offer potential for reducing melanin production. Elucidating these endocrine mechanisms provides essential insights for developing more effective and personalized therapeutic strategies for melasma.

## 1. Introduction

Melasma is a prevalent acquired hyperpigmentary disorder characterized clinically by symmetrical, hyperpigmented macules or patches, which most commonly occur on sun-exposed areas of the face. These visible manifestations underscore its significance as one of the most prevalent cutaneous conditions [1], particularly in women and individuals with darker skin types. Molecular studies have identified nearly 300 genes differentially expressed in melasma-affected skin, highlighting its remarkable biological complexity and multifactorial etiology [2]. Clinically, the disorder presents with confluent brownish macules of varying intensity, defined by irregular yet sharply demarcated borders, which often impart a ‘stuck-on’ appearance [3]. From a pathological perspective, hypermelanosis in melasma is subclassified into epidermal, dermal, and mixed types, with brownish, bluish-grey, or combined discoloration patterns, respectively [4]. Although reported across diverse ethnic populations, melasma is prevalent in individuals with Fitzpatrick skin types III to VI [5,6]. Epidemiological investigations have consistently identified ultraviolet (UV) radiation as the predominant environmental trigger, implicated in up to 50% of cases, with additional roles played by hormonal fluctuations, genetic susceptibility, and other endocrine disturbances [7,8]. UV exposure activates melanogenesis in melanocytes and induces paracrine effects on keratinocytes, fibroblasts, and mast cells, promoting the release of melanogenic mediators such as alpha-melanocyte-stimulating hormone (alpha-MSH), corticotropin, and SCF (Stem cell factor) [9]. In 2018, Regazzetti et al. proposed that opsin-3 (OPN3) might be a key receptor responsible for visible light-induced hyperpigmentation [10]. Genetic factors, including melanocortin-1 receptor (MC1R), solute carrier family 24 member 5 (SLC24A5), tyrosinase (TYR), and oculocutaneous albinism II (OCA2), significantly impact pigmentation [11]. Histopathology of melasma revealed increased melanin in the suprabasal and basal layers (100%), melanophages in the upper dermis, and solar elastosis (65%) in contrast to facial pigmentary demarcation lines, wherein increased basilar melanin (75%) and dermal melanophages were the key findings [12]. These findings indicate that melasma arises from a complex interplay of intrinsic and extrinsic factors, wherein endocrine influences—particularly sex hormones—emerge as indispensable determinants of disease susceptibility and chronicity.

Hormonal regulation has therefore become a central focus in melasma research. Estrogen, beyond its established roles in female reproductive maturation, secondary sexual trait development, and pubertal growth, exerts profound effects on cutaneous pigmentation [13]. Progesterone, predominantly synthesized in the corpus luteum and placenta, contributes to reproductive regulation and pigmentary modulation [14,15,16]. Epidemiological data reveal melasma in 14.5–56% of pregnant women and 11.3–46% of oral contraceptive users, with prevalence varying by geographic and ethnic factors [17]. Intriguingly, a multinational cohort of 324 women demonstrated that only 20% of cases developed during pregnancy, while approximately 10% arose post-menopause, further implicating dynamic hormonal influences in disease onset [18]. Mechanistic studies demonstrate that melanocytes in melasma lesions display increased expression of estrogen and other sex hormone receptors, with heightened sensitivity to hormonal stimulation, thereby accelerating pigment production and recurrence [19]. Despite the well-established clinical association, the molecular pathways through which estrogen and progesterone dysregulate melanogenesis remain incompletely understood and often contradictory. Elucidating these hormones’ dual genomic and non-genomic roles, their crosstalk with oxidative stress and inflammation, and their regulatory effects on melanogenic enzymes is therefore critical. This review synthesizes current knowledge on the endocrine dimension of melasma pathogenesis while exploring emerging hormone-targeted therapeutic strategies. By bridging mechanistic insights with clinical applications, such an approach aims to inform the development of more effective, individualized interventions for this challenging and recurrent condition.

## 2. Mechanisms of Estrogen and Progesterone in Melasma

### 2.1. Mechanism of Estrogen’s Effect on Melasma

Estrogens, particularly estradiol (E2), estrone (E1), and estriol (E3), exhibit distinct molecular structures that influence their biological activity and bioavailability. Estradiol, recognized as the most biologically active form, possesses a 17β-hydroxyl group that enhances its affinity for estrogen receptors, thereby facilitating its potent physiological effects [20]. Estrone functions as a metabolic precursor to estradiol, with potential for interconversion between these forms; enzyme 17-hydroxysteroid dehydrogenase (17-HSD) facilitates this process [21]. In contrast, estrone can be converted into estradiol, and the two can also be interconverted by the enzyme 17β-HSD [22]. The distinct molecular structures of different estrogens not only affect how they bind to receptors and how they are also shape their subsequent biological effects [23]. This is especially important in understanding their clinical applications, particularly in therapies and diseases involving estrogen. Estrogen plays a multifaceted role in the physiological processes of the skin, influencing aspects such as pigmentation, hydration, elasticity, and overall appearance. It regulates moisture levels, enhances skin elasticity, and promotes an even skin tone, contributing to the skin’s health and vitality. Among the various forms of estrogen—E1, E2, and E3—each exhibits distinct biological activities and follows unique metabolic pathways, affecting melanocytes, the cells responsible for pigment production. Their involvement in conditions such as melasma has garnered increasing research attention. Specifically, E2, E1, and E3 influence the development and function of melanocytes through various mechanisms, highlighting the complex relationship between estrogen and skin health [24]. Understanding these variations is crucial for gaining insights into the underlying pathophysiological mechanisms of melasma and maintaining a balance between these hormones.

#### 2.1.1. Estrogen Receptor-Mediated Melanin Production: The Dual Pathway Role of Estradiol (E2) in the Pathogenesis of Melasma

E2 is a steroid hormone composed of ten carbon atoms, primarily synthesized from androstenedione, and functions as the most potent estrogen among the three types of estrogen: E1, E2, and E3. The biological activity of these estrogens is in a ratio of 10:5:1 for E2:E1:E3, with estradiol being twelve times more potent and eight times more effective in its biological functions than the other estrogens [25,26]. E2 is synthesized from progesterone through the intermediate pregnenolone, derived from cholesterol, and acts as an agonist for estrogen receptors (ERs) [27,28]. These receptors, estrogen receptors-α (ER-α) and estrogen receptors-β (ER-β), are nuclear hormone receptors that function as ligand-activated transcription factors [29]. Upon binding to E2, they undergo conformational changes, allowing them to interact with estrogen response elements (EREs) in target gene promoters, modulating transcriptional activity [30,31]. Additionally, estradiol interacts with G-protein-coupled estrogen receptors (GPER) at the cell membrane, initiating rapid non-genomic effects [32,33]. GPER, exhibiting high selectivity for E2, triggers signaling cascades that involve intracellular calcium ion (Ca^2+^) increases, cyclic adenosine monophosphate (cAMP), and protein kinases. These pathways regulate downstream transcription factors and epigenetic regulators, influencing cellular responses. The effects of estrogen on the skin are likely mediated through both genomic (nuclear) and non-genomic (membrane) signaling pathways. Unlike classical ERs, GPER has a much higher affinity for E2, significantly reducing binding to other endogenous estrogens like E1 and E3 [34,35] (Figure 1).

E2 plays a significant and complex role in regulating melanin production and skin pigmentation [36]. As the most potent endogenous estrogen, it enhances melanogenesis by binding to and activating nuclear estrogen receptors (ER-α and ER-β) and the membrane-associated GPER. This dual activation triggers a cascade of signaling pathways. Through its genomic actions, the E2–ER complex binds to estrogen response elements (EREs) located in the promoter regions of target genes, thereby promoting the transcriptional upregulation of key melanogenic enzymes, including TYR, tyrosinase-related protein 1 (TRP-1), and tyrosinase-related protein 2 (TRP-2) [37]. Concurrently, non-genomic signaling via GPER rapidly activates secondary messengers like cAMP and Ca^2+^, further modulating melanogenic activity and melanocyte proliferation. This hormonal influence explains the frequent onset or worsening of hyperpigmentation disorders like melasma during periods of elevated estrogen, such as pregnancy or hormone therapy, as evidenced by the increased expression of estrogen receptors in melasma-affected skin [38]. These processes contribute to the prevention of skin ageing by increasing skin thickness, reducing wrinkles, and improving moisture retention. Consequently, the relative hypoestrogenism accompanying menopause aggravates the adverse effects of both intrinsic and extrinsic ageing factors [39].

While estrogen provides protective benefits against skin ageing, it paradoxically promotes skin pigmentation by enhancing the activity of pigment-producing cells. Ranson et al. [40]. Incubation with β-estradiol was first shown to enhance tyrosinase activity in neonatal melanocytes dose-dependently, and later studies confirmed that 17β-estradiol also modulates melanogenesis and mitogenesis in cultured human neonatal melanocytes [41]. Notably, Jee et al. reported that, following the addition of 17β-estradiol to melanocyte cultures, there was a dose-dependent increase in melanocyte proliferation. This was accompanied by a reduction in both tyrosinase activity and melanin content [36]. Maeda et al. demonstrated that, upon incubation with ovarian hormones (E2, E3, and progesterone), there was a similar pattern of melanocyte proliferation without an elevation in tyrosinase activity [42]. The results reported by Jee et al. contrast the findings from two distinct studies conducted by the same Australian research group. In these latter studies, in vitro-cultured melanocytes derived from normal human skin were incubated with β-estradiol at physiological concentrations comparable to those observed during pregnancy. This incubation resulted in a dose-dependent enhancement of tyrosinase activity, increased melanin secretion and a concomitant reduction of approximately 50% in melanocyte numbers. [38,43]. In 1998, Kippenberger et al. demonstrated an upregulation of tyrosinase activity, along with increased expression of TRP-1 and TRP-2, in melanocytes that were isolated from the back skin of human fetuses and subsequently cultured with estradiol [44]. A comparative analysis of the expression levels of ER-α and ER-β, alongside progesterone receptors, between melasma-affected lesions and adjacent healthy skin revealed an upregulation of these receptor proteins [45]. This finding is further supported at the cellular level by ER-β on melanocytes. At the molecular level, it is reinforced by the induction and upregulation of TYR, which acts as the pivotal enzyme in the rate-limiting step of melanin production [46]. These results suggest that E2 plays a significant role in the pathogenesis of melasma.

#### 2.1.2. Estrone (E1) in the Pathogenesis of Melasma: Mechanisms of Upregulating Melanogenic Enzymes and Modulating Melanocyte Behavior

E1 is a C18 steroid hormone, specifically classified as a 3-hydroxy-1,3,5(10)-estratriene-17-one, which plays a crucial role in various biological processes. Its intricate chemical structure is characterized by a phenolic A-ring, which contributes to its unique properties, a cyclopentane D-ring that adds to its stability, and a ketone at the C17 position, which distinctly differentiates it from other estrogens such as E2. This specific arrangement of rings and functional groups defines its classification and influences its interactions within the body, impacting a range of physiological functions and signaling pathways [47]. E1 is primarily synthesized in the ovaries, placenta, and adipose tissue, with the ovaries being the predominant source during reproductive years. In postmenopausal women, estrone becomes the primary circulating estrogen, produced mainly through the aromatization of androgens such as androstenedione and testosterone via the enzyme aromatase [48,49]. This conversion process is critical, as it allows for the maintenance of estrogen levels in women who no longer produce significant amounts of estradiol from the ovaries. The biosynthesis of E1 is also influenced by peripheral tissues, particularly adipose tissue, which can convert androgens into estrogens, thus playing a significant role in the estrogenic milieu of postmenopausal women [50].

During the process of melanin formation, E1 has a significant impact on the expression of key enzymes (especially TYR) and tyrosinase-related proteins(TRP-1 and TRP-2) in the melanin biosynthesis pathway [51,52]. The conversion of L-tyrosine to melanin is highly dependent on these core enzymes. TYR plays a central catalytic role and has an essential regulatory significance for melanin production rate. Studies have shown that E1 can upregulate the expression of these key enzymes, and the enhanced enzyme activity will further promote melanin synthesis. The potential mechanism of this process involves the activation of multiple key signaling pathways, including the cAMP/PKA MAPK pathway and the Wnt/β-catenin pathway, both of which support melanocyte function and melanin production [53,54,55]. Specifically, E1 can activate the cAMP/PKA signaling pathway, promoting the phosphorylation of transcription factors, thereby upregulating the expression of TYR and other melanin production-related proteins; at the same time, E1 can also regulate the transcription factors related to microphthalmia (MITF)—this factor plays a central role in the development and functional regulation of melanocytes. The induction of MITF expression by E1 may initiate a downstream cascade reaction, stimulating the transcription of key enzymes involved in melanin synthesis to promote melanin production [56,57,58]. Additionally, the influence of E1 on melanocyte function is more complex due to its potential interaction with the Wnt/β-catenin pathway: the Wnt/β-catenin pathway is crucial for cell differentiation, proliferation, and survival regulation, especially for melanocytes. E1 may affect melanocytes’ differentiation process and functional capacity by regulating this pathway. It is well established that the Wnt signaling pathway is crucial in regulating melanocyte development and the pigmentation process. Abnormal regulation of this pathway can lead to abnormal melanocyte activity, potentially inducing pigmentation disorders such as melasma [59].

The relationship between estrone and melanocyte behavior is remarkably complex, intricately influencing not only the skin pigmentation that determines the color and tone of our skin but also the dynamic cellular processes that govern the growth, development, and function of these specialized pigment-producing cells. At lower concentrations, E1 has been shown to stimulate melanocyte proliferation, supporting the maintenance of these pigment-producing cells in the skin. This effect is partly mediated through the modulation of key cell cycle regulators such as cyclin D1 and p21, which facilitates progression from the G1 to S phase. In contrast, at elevated concentrations, E1 may suppress melanocyte growth, potentially leading to reduced viability and changes in pigmentation [60,61,62]. E1 enhances the ability of melanocytes to migrate, which is crucial for the distribution of pigment across the epidermis. The enhanced migration ability may play a significant role in the onset and dissemination of pigmentation disorders, such as melasma [63,64,65]. The intricate balance between cell proliferation and migration highlights the complexity of E1’s function in melanocyte biology.

#### 2.1.3. Beyond a Weak Estrogen: Unraveling Estriol (E3) Pro-Pigmentary Effects in Melasma

E3 is a naturally occurring form of estrogen produced primarily during pregnancy. Its concentration increases substantially due to intricate physiological exchanges between the placenta and developing fetal systems [66]. E3 is mainly synthesized through a metabolic pathway that begins with the conversion of dehydroepiandrosterone (DHEA) into E1. This step is followed by further hydroxylation reactions that convert E1 into E3. These processes are facilitated by specific enzymes, including aromatase and 17β-hydroxysteroid dehydrogenase, which play essential roles in the overall biosynthesis [67]. Unlike E2, which is well known for its strong estrogenic effects, E3 binds to estrogen receptors with much lower affinity. Because of this, it has often been regarded as a milder estrogen. E3’s distinct metabolic profile and ability to interact with other hormones indicate it may still play specialized roles—especially in the skin. Studies have shown that E3 can influence skin function by modulating key processes like cell proliferation and differentiation, which are essential for keeping skin healthy and balanced [68]. The differences in metabolism between E3 and other estrogens, particularly E2, suggest that E3 may have unique effects on skin tissue. This could be due to its specific receptor interactions or ability to activate different signaling pathways compared to more potent estrogens [69,70].

The relationship between E3 and oxidative stress is critically important for understanding the pathogenesis of melasma. Emerging lines of evidence indicate that E3 exerts a critical regulatory function in oxidative stress homeostasis, primarily through the transcriptional upregulation of pivotal antioxidant enzymes, including superoxide dismutase (SOD) and glutathione peroxidase (GPx). These enzymes serve pivotal functions in cellular defense against oxidative stress, catalyzing the dismutation of superoxide anions into hydrogen peroxide and molecular oxygen, and further mediating the decomposition of hydrogen peroxide into water and oxygen. Through upregulation of their activity, E3 modulates the production of reactive oxygen species (ROS), which are key mediators of oxidative stress. Excessive ROS levels can induce oxidative injury in skin cells, triggering a cascade of signaling events that stimulate melanogenesis—the process underlying melanin synthesis. In addition to the direct effects on enzymatic activity, oxidative stress is also known to activate several signaling cascades that contribute to the overall enhancement of melanogenesis. For example, the nuclear factor kappa-light-chain-enhancer of activated B cells (NF-κB) pathway is activated under oxidative stress conditions. It has been implicated in the regulation of genes associated with melanin production. Activating NF-κB leads to the transcription of pro-inflammatory cytokines, which can further stimulate melanocyte activity and increase melanin synthesis. This mechanism is especially relevant in melasma, a common hyperpigmentation disorder where oxidative stress is thought to worsen pigmentation. E3 may enhance ROS generation through various pathways, including altering the body’s antioxidant defense system and directly interfering with cellular signaling processes that control ROS levels [71,72,73,74].

E3 are integral regulators in cellular signaling pathways, particularly in activating the extracellular signal-regulated kinase (ERK) pathway. This activation leads to a cascade of downstream signals that result in the transcriptional upregulation of genes involved in the cellular response to oxidative stress [75,76]. The balance between oxidative stress and these signaling pathways is pivotal in the pathogenesis of various conditions, including melasma. ROS, generated by oxidative stress, can trigger melanogenesis pathways, increasing melanin production. This enhanced melanogenesis contributes to the development of melasma. ROS induce melanogenesis and promote the expression of oxidative stress-related genes, establishing a feedback loop that exacerbates pigmentation. Understanding how E3 influences oxidative stress responses and their role in melanogenesis could open up novel therapeutic avenues for melasma. Targeting specific E3 within these pathways may provide potential biomarkers or therapeutic targets for effective management of melasma, potentially restoring the balance between oxidative stress and pigmentation processes.

### 2.2. The Mechanism of Progesterone’s Effect on Melasma

Progesterone, a steroid synthesized from cholesterol via pregnenolone in the corpus luteum, placenta, and adrenal cortex, regulates the menstrual cycle and pregnancy. Its metabolite, 20α-hydroxyprogesterone (20α-DHP), formed by 20α-hydroxysteroid dehydrogenase (20α-HSD), exhibits distinct biological activities in reproduction, metabolism, and the central nervous system as a neurosteroid. Both progesterone and 20α-DHP upregulate melanogenic enzymes (TYR, TRP-1, TRP-2) and activate cAMP/PKA and MAPK pathways, enhancing melanin synthesis. Additionally, they promote melanocyte proliferation and migration at low concentrations, influencing extracellular remodeling and contributing significantly to melasma hyperpigmentation [77,78].

#### 2.2.1. The Role of Progesterone in Melasma: From Hormonal Fluctuations to Pigmentation Disorders

Progesterone was among the earliest hormones to be characterized. Often recognized alongside estrogen, it is widely classified as a female sex steroid [79]. Progesterone, an endogenous 21-carbon steroid hormone, is biosynthesized from cholesterol via pregnenolone. It represents a key gonadal hormone. In non-pregnant individuals, progesterone is primarily synthesized in the corpus luteum of the ovaries. During pregnancy, in addition to ovarian production in the early stages, the placenta becomes a primary site of progesterone synthesis, playing essential roles in maintaining pregnancy and regulating various physiological processes related to reproduction. To a relatively minor degree, progesterone is synthesized in smaller quantities by the adrenal cortex. In males, the Leydig cells within the testes also produce progesterone at low levels. Additionally, adipose tissue and specific other tissues contribute to progesterone production, albeit in minute amounts [14,15,80]. Like certain steroids, progesterone is synthesized within the nervous system by both neurons and glial cells, a process known as neurosteroidogenesis. Moreover, progesterone affects nervous system tissues, a phenomenon referred to as neuroactive steroid action. The enzymes that are essential for the transformation of cholesterol into pregnenolone and further into progesterone are extensively found in the brain [81]. Progesterone is subject to further metabolic processes that yield other neuroactive steroids. Among these, allopregnanolone holds the most tremendous significance [15,82].

Progesterone contributes to the pathogenesis of melasma through multiple interconnected mechanisms that dysregulate melanogenesis, promote vascular remodeling, and induce a pro-inflammatory microenvironment. At the molecular level, progesterone binding to nuclear or membrane receptors activates the PI3K/Akt/GSK3β pathway, leading to phosphorylation of Akt and inhibition of GSK3β [53,54]. This activation leads to the downstream phosphorylation of transcription factors, including cAMP response element-binding protein (CREB), which relocates to the nucleus to upregulate the expression of critical melanogenic enzymes such as TYR, TRP-1, and TRP-2. Simultaneously, progesterone induces oxidative stress by elevating lipid peroxidation markers (e.g., MDA) and diminishing antioxidant enzyme activity (e.g., SOD), thereby disrupting redox balance and further stimulating melanogenesis via pathways like Nrf2/Keap1 [83,84]. In the tissue microenvironment, progesterone enhances angiogenesis by upregulating vascular endothelial growth factor (VEGF), facilitating increased vascular density and inflammatory cell infiltration [85]. Inflammatory cytokines such as interleukin-6 (IL-6) and TNF-alpha (TNF-α) subsequently stimulate melanocyte proliferation and melanin production, reinforcing hyperpigmentation. Additionally, crosstalk between progesterone signaling and core transcriptional regulators, such as MITF and beta-catenin, integrates upstream inputs, amplifying melanogenic output [86,87,88].

The role of progesterone in the pathogenesis of melasma is increasingly understood through the investigation of its receptor subtypes and the associated downstream signaling pathways [89]. Recent research suggests that progesterone receptor-A(PR-A) might exert a more significant impact on the development of melasma compared to progesterone receptor-B(PR-B), particularly during hormonal fluctuations linked to the menstrual cycle. These hormonal changes can exacerbate pigmentation disorders. When these receptors are activated, they regulate a variety of downstream effectors, including transcription factors vital in controlling melanogenesis [45,90,91]. The binding of progesterone to its receptors can augment the expression of the MITF, which is essential for regulating the activity of melanocytes and the synthesis of melanin. This receptor-mediated signaling pathway causes an elevation in the activity of tyrosinase and other enzymes involved in melanogenesis, ultimately giving rise to the hyperpigmentation typically linked to melasma [92,93].

Epigenetic modifications and non-coding RNAs are increasingly acknowledged as crucial contributors to the molecular mechanisms implicated in melasma, particularly regarding the signaling pathways of progesterone [94]. Epigenetic alterations, such as DNA methylation and histone modification, can modulate gene expression without changing the DNA sequence, thus influencing the activity of genes associated with melanin production [95,96]. The methylation status of the promoter regions of genes involved in melanogenesis can be affected by hormonal signaling, such as that from progesterone. This suggests that fluctuations in hormone levels may lead to reversible epigenetic modifications, potentially raising the probability of melasma development in individuals [97]. Non-coding RNAs, especially microRNAs (miRNAs), are crucial in regulating gene expression in melanocytes. Specific miRNAs can influence the levels of vital proteins essential for melanin synthesis, including tyrosinase and MITF. They achieve this by binding to the mRNA transcripts of these proteins, which inhibits their translation into functional proteins. For example, miR-145 and miR-199 have been demonstrated to decrease the expression of MITF, leading to a reduction in melanin production when hormonal signals are present [98,99,100]. The interaction between progesterone, epigenetic modifications, and non-coding RNAs reveals a sophisticated regulatory network (Figure 2).

#### 2.2.2. From Hormone to Pigment: 20α-DHP as the Link Between Steroid Metabolism and Skin Pigmentation

20α-DHP is a steroid hormone classified under the C_21_ steroid category, with a molecular formula of C_21_H_30_O_3_. Its structural uniqueness arises from introducing an α-oriented hydroxyl group at the C-20 position of the progesterone backbone. This modification significantly alters its polarity and biological activity compared to its stereoisomer, 20β-hydroxyprogesterone (20β-DHP) [101,102]. 20α-DHP, chemically designated as 20α-hydroxy-4-pregnene-3,20-dione, is an endogenous steroid hormone primarily synthesized from progesterone through the action of the enzyme 20α-hydroxysteroid dehydrogenase (20α-HSD) [103]. This enzymatic conversion occurs predominantly in the ovaries, placenta, and adrenal glands, where progesterone is metabolized to yield 20α-DHP. The synthesis of 20α-DHP is a critical aspect of steroidogenesis, reflecting the complex interplay of various steroidogenic enzymes and pathways [102]. Following its formation, 20α-DHP undergoes further metabolic transformations in the liver, facilitating its eventual excretion via urine. Despite its relatively weak progestogenic activity compared to its precursor, progesterone, emerging studies suggest that 20α-DHP may exert biological effects through non-classical receptor pathways [104]. The role of 20α-DHP in regulating the MITF and MC1R signaling pathways is essential for understanding how it affects melanin production. MITF is a key transcription factor that controls melanogenesis—the process of melanin synthesis. Meanwhile, MC1R is an upstream regulator influencing MITF activity [105].

The regulation of melanin synthesis relies significantly on the interaction mediated by 20α-DHP, which appears to influence both MITF expression and signaling pathways that enhance melanin production. Studying these interactions can help clarify the biological mechanisms behind pigmentation, and their potential relevance to skin color-related conditions and melanoma [106,107]. Research suggests that 20α-DHP may exert its effects by binding to PR or via non-genomic mechanisms, activating the MC1R-MITF axis [108]. An activation process drives increased expression of downstream pigment-related genes such as TYR and TYR1. Both genes play essential roles in melanin biosynthesis, underscoring their importance in biological pigmentation. Research using animal models and cell cultures has shown that treatment with 20α-DHP promotes enhanced translocation of MITF into the nucleus, leading to increased melanin production [109,110,111]. Under normal physiological conditions, 20α-DHP is essential for regulating the menstrual cycle, supporting pregnancy, and maintaining a balance of steroid hormones. This steroid hormone is found in various tissues, including the skin, where it can influence biological processes through local production or circulating to reach target cells [112]. In skin physiology, 20α-DHP may play a significant role in influencing pigmentation processes, including the production and distribution of melanin. These processes are crucial in the development of skin conditions like melasma. Notably, during pregnancy, elevated levels of 20α-DHP have been observed, suggesting a potential link to the onset of melasma, commonly seen in pregnant women [113,114]. However, the specific mechanisms by which 20α-DHP contributes to pigmentation changes remain to be elucidated, warranting further investigation into its role in skin health and disease (Table 1).

## 3. Treatment Strategies Based on the Hormones of Both

### 3.1. Treatment Strategies for Melasma Based on Estrogen

Estrogen modulators, particularly selective estrogen receptor modulators (SERMs) such as raloxifene and bazedoxifene, have shown promising potential in treating melasma, a common skin condition characterized by hyperpigmentation. These agents leverage their tissue-selective actions to modulate estrogen receptors in the skin, reducing hyperpigmentation while supporting skin health without systemic side effects. With its established safety profile, Raloxifene and bazedoxifene, known for its dual estrogenic and anti-estrogenic effects, have demonstrated efficacy in clinical trials, significantly reducing the severity of melasma lesions compared to baseline measurements. The mechanism of action is believed to involve the modulation of estrogen receptors in the skin, which can influence melanocyte activity and melanin synthesis. For example, a randomized controlled trial indicated that patients receiving SERMs experienced a marked improvement in pigmentation scores after 12 weeks of treatment. This highlights the potential of these agents as a viable therapeutic option for melasma management, addressing both aesthetic concerns and underlying hormonal factors. However, the long-term effects of SERMs in treating melasma remain an area of active investigation. While current studies suggest that SERMs, which act as estrogen receptor agonists or antagonists depending on the tissue type, may offer benefits by modulating hormonal pathways that influence skin pigmentation, the long-term safety and efficacy need thorough evaluation. Research indicates that estrogen plays a significant role in skin health, including pigmentation and elasticity, and that SERMs could potentially mitigate the hyperpigmentation associated with melasma [115,116,117,118].

Additionally, aromatase inhibitors represent a promising therapeutic approach for alleviating melasma symptoms through their ability to lower systemic estrogen levels significantly. By reducing circulating estrogen concentrations, these inhibitors effectively decrease the stimulation of melanocytes, which are known to be highly responsive to estrogenic activity. However, this treatment modality requires careful clinical management and monitoring, as excessive estrogen suppression may lead to undesirable dermatological side effects such as skin dryness, thinning, or even atrophy. Beyond direct estrogen receptor targeting, emerging research suggests that strategic modulation of estrogen metabolism pathways could benefit melasma patients. Specifically, interventions aimed at improving the metabolic ratio between 2-hydroxyestrone (2-OHE1) and 16-alpha-hydroxyestrone (16α-OHE1)—whether through dietary modifications, pharmacological agents, or targeted enzymatic inhibition of CYP1A1 and CYP1B1—may further reduce hyperpigmentation by promoting the production of less biologically active estrogen metabolites [119,120,121]. The therapeutic potential of these approaches could be substantially enhanced when combined with antioxidant therapies, as oxidative stress is recognized as a significant contributor to melanocyte dysfunction. This integrated treatment strategy, combining hormonal regulation, metabolic modulation, and oxidative stress reduction, offers a comprehensive and scientifically grounded multi-targeted approach for managing the complex pathophysiology of melasma, potentially yielding superior clinical outcomes compared to single-mechanism interventions.

### 3.2. Treatment Strategies for Melasma Based on Progesterone

The development of selective progesterone receptor modulators (SPRMs) significantly advances local treatment strategies for conditions like melasma [122]. These agents can selectively modulate the activity of progesterone receptors in the skin, potentially reducing the hyperpigmentation associated with excessive progesterone activity. Research has demonstrated that SPRMs can effectively counteract the stimulatory effects of progesterone on melanocyte activity, reducing melanin production [123]. One of the primary mechanisms through which SPRMs exert their effects is by inhibiting progesterone-induced tyrosinase activity and the subsequent melanin synthesis pathways. TYR, a key enzyme in the melanin biosynthesis pathway, catalyzes the conversion of tyrosine to dihydroxyphenylalanine (DOPA) and subsequently to DOPA quinone, which are critical steps in melanin production. In a study investigating the effects of asoprisnil (AP), a specific SPRM, it was found that AP effectively suppressed extracellular melanin levels in B16F10 mouse melanoma cells, demonstrating its potential to inhibit melanin production at concentrations that were nontoxic to the cells [124,125]. Treatment strategies employing 20-hydroxyprogesterone analogues in the localized management of melasma establish a groundbreaking paradigm, leveraging the unique properties inherent to this metabolite (Table 2).

Distinct from the conventional biochemical activities attributed to endogenous progesterone, the steroidal derivative 20α-DHP demonstrates a broader spectrum of pharmacological actions, particularly characterized by pronounced anti-inflammatory and anti-proliferative properties [126]. Such biological activities render this compound a promising candidate for dermatological interventions, especially in conditions characterized by dysregulated melanogenesis and chronic inflammation, such as melasma and other hyperpigmented dermatoses [127]. Mechanistic investigations suggest that 20α-DHP can modulate key inflammatory signaling cascades, thereby attenuating the recruitment and activation of pro-inflammatory mediators within cutaneous tissues. Concurrently, its inhibitory influence on melanocyte metabolic activity and proliferative dynamics underscores its therapeutic potential to normalize pigmentation pathways, directly addressing pivotal etiological determinants in the pathophysiology of pigmentary disorders. From a translational perspective, future research trajectories should be directed toward the rational design and refinement of topical or transdermal formulations incorporating 20α-DHP. Particular emphasis ought to be placed on optimizing drug delivery systems, including nanocarriers, liposomal encapsulation, and penetration enhancers, to facilitate superior dermal absorption and targeted bioavailability. Such advancements would augment penetrative efficacy and enhance therapeutic potency, ultimately fostering more effective and sustained clinical outcomes in the management of hyperpigmentation disorders (Figure 3).

## 4. Conclusions and Prospects

Current evidence clearly demonstrates the pivotal roles of estrogen and progesterone in the onset and progression of melasma. These hormones regulate melanogenesis not only through genomic and non-genomic signaling pathways that directly modulate melanogenic enzymes (e.g., TYR, TRP-1, TRP-2) and transcription factors (e.g., MITF) but also by interacting with oxidative stress, inflammatory responses, and vascular remodeling, thereby establishing a complex molecular network that exacerbates hyperpigmentation. The differential receptor affinities and metabolic pathways of the three estrogen subtypes (E1, E2, E3) contribute to the heterogeneity of their effects on melanocytes. At the same time, progesterone and its metabolites (e.g., 20α-DHP) amplify melanogenic signaling through multiple downstream pathways. Such mechanistic diversity provides essential insights into the origins of distinct clinical phenotypes and individual susceptibility. Although hormonal therapy and molecular targeted therapy have shown promising prospects, localized interventions such as hydroquinone, tranexamic acid, and Q-switched laser therapy remain the standard treatment modalities for melasma [128]. The combination of these agents with hormonal therapy not only directly suppresses melanogenesis but also mitigates pigmentation while reducing treatment-associated adverse effects. For instance, hydroquinone competitively inhibits tyrosinase activity by interfering with tyrosine utilization, thereby blocking melanin synthesis [129]. Tranexamic acid has also angiogenesis induced by VEGF and ET-1, further improving hyperpigmentation [130]. An integrated approach should be adopted to achieve optimal therapeutic outcomes, encompassing hormonal modulation, molecular targeted interventions, and conventional clinical therapies. Specifically, hormonal regulation can be achieved using selective hormone receptor modulators to modulate estrogen-mediated melanogenic pathways; topical treatment involves the application of agents such as hydroquinone or tranexamic acid to interrupt the melanogenesis process directly; photoprotection measures are essential to prevent UV-induced skin damage throughout the treatment course. Therefore, a multidimensional combined treatment strategy is necessary to enhance clinical efficacy in melasma management.

Future investigations should concentrate on several directions. First, multi-omics approaches, including transcriptomics, epigenomics, and single-cell sequencing, are necessary to elucidate the dynamic regulation of hormonal signaling and melanogenesis, thereby uncovering the mechanisms underlying individual predisposition. Second, more attention should be directed toward the roles of non-coding RNAs and epigenetic modifications in hormone-mediated remodeling of melanocyte function, which may help reconcile contradictory findings and offer novel molecular targets. At the translational level, rigorous evaluation of the long-term efficacy and safety of selective receptor modulators (SERMs, SPRMs), aromatase inhibitors, and their combination with antioxidant therapies is essential. Ultimately, integrating basic mechanistic studies with precision medicine strategies may facilitate the development of personalized hormonal interventions. However, the current evidence for SERMs, SPRMs, and aromatase inhibitors in melasma is preliminary, and further long-term, large-scale clinical trials are required before these approaches can be translated into standard care.

## Figures and Tables

**Figure 1 ijms-26-10856-f001:**
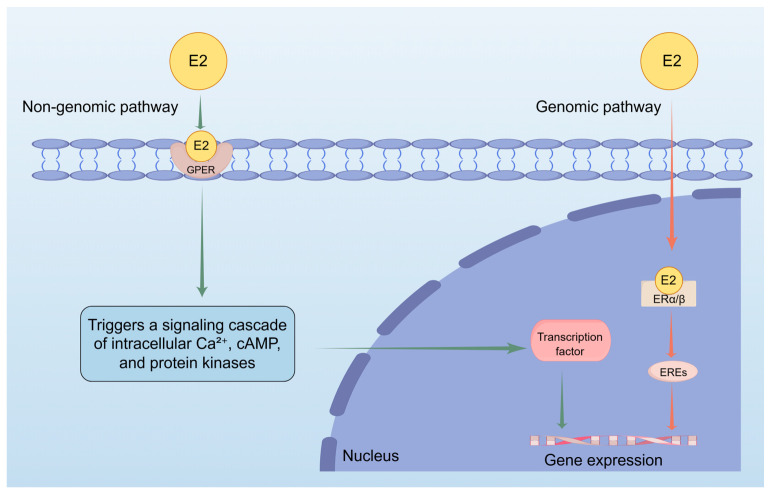
This figure shows how estradiol (E2) regulates gene expression through two pathways. Non-genomic pathway: Estradiol (E2) interacts with G protein-coupled estrogen receptor (GPER) at the cell membrane, initiating a series of signal transduction processes involving intracellular calcium (Ca^2+^) increase, cyclic adenosine monophosphate (cAMP) and protein kinases. These pathways would regulate downstream transcription factors, thereby influencing cellular responses and promoting gene expression. Genomic pathway: When E2 enters the nucleus and binds to nuclear receptors, it activates estrogen response elements (EREs) and promotes gene expression.

**Figure 2 ijms-26-10856-f002:**
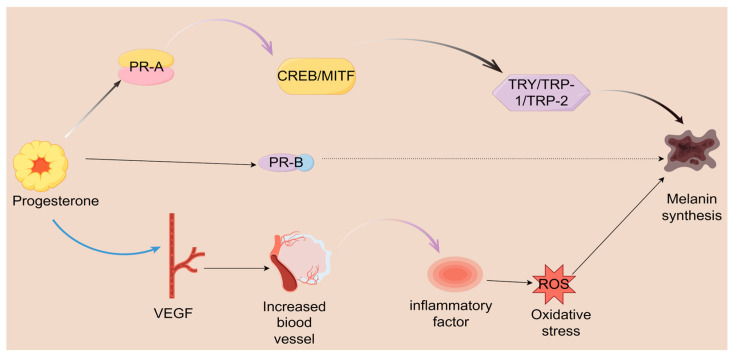
Progesterone is crucial in regulating melanin production and various physiological processes through a complex mechanism. It binds to two types of receptors, PR-A and PR-B. Upon binding to the PR-A receptor, progesterone activates a cascade of events that includes the activation of CREB and MITF. This activation subsequently regulates the expression of key enzymes involved in melanin synthesis, specifically TYR TRP-1 and TRP-2, ultimately promoting the formation of melanin. VEGF increases vascularization and generates inflammation, leading to oxidative stress, ultimately affecting melanin synthesis.

**Figure 3 ijms-26-10856-f003:**
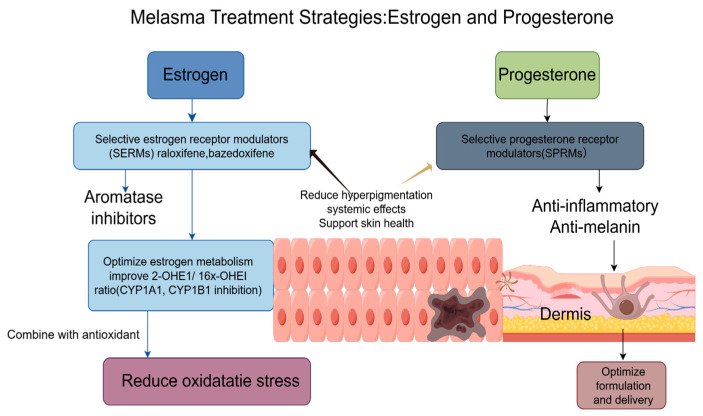
Melasma treatment strategies that focus on estrogen and progesterone pathways involve a multifaceted therapeutic approach. This incorporates using SERMs, such as raloxifene and bazedoxifene, as well as topical aromatase inhibitors that help reduce hyperpigmentation while avoiding systemic effects. Additionally, anti-inflammatory agents help optimize estrogen metabolism by influencing the balance between 2-OHE1 and 16X-OHE1 by inhibiting CYP1A1 and CYP1B1 enzymes. Within the dermis, SPRMs work to suppress progesterone-stimulated melanin production directly. Combining these approaches with antioxidants further supports treatment efficacy by reducing oxidative stress and enhancing overall results.

**Table 1 ijms-26-10856-t001:** Comparative roles of estrogen and progesterone in melanogenesis and melasma pathogenesis (↑: Increased levels).

Hormone	Main Source(s)	Receptor(s) Involved	Key Signaling Pathways Activated	Effects on Melanogenesis	Additional Contributions (Oxidative Stress, Inflammation, Vascularization)
Estradiol (E2)	Ovaries, peripheral conversion	ER-α, ER-β, GPER	Genomic (ERE-mediated), Non-genomic (MAPK, cAMP, Ca^2+^)	↑ TYR, TRP-1/2 expression; ↑ melanocyte proliferation; ↑ melanin deposition	Enhances skin thickness & moisture; paradoxically promotes hyperpigmentation
Estrone (E1)	Ovaries, adipose tissue (postmenopause)	ER-α, ER-β	cAMP/PKA, MAPK, MITF	↑ TYR, TRP-1/2 expression; concentration-dependent effects on proliferation/migration	Modulates cell cycle (Cyclin D1, p21); ↑ melanocyte migration
Estriol (E3)	Placenta (during pregnancy)	ER (low affinity)	NF-κB, ROS-related pathways	Mild estrogenic effect; modulates proliferation/differentiation	↑ ROS generation, aggravates oxidative stress contributing to melasma
Progesterone	Corpus luteum, placenta, adrenal cortex	PR-A, PR-B, membrane PRs	PI3K/Akt/GSK3β, CREB/MITF, Nrf2/Keap1	↑ TYR, TRP-1/2 expression; ↑ melanocyte proliferation & melanin synthesis	↑ VEGF-mediated angiogenesis; ↑ IL-6/TNF-α inflammation; redox imbalance
20α-DHP	Progesterone metabolite (ovary, placenta)	PR, MC1R-MITF axis	MITF nuclear translocation, cAMP/PKA	↑ TYR, TYR1 expression; ↑ melanin biosynthesis	Pregnancy-associated elevation linked to melasma onset

**Table 2 ijms-26-10856-t002:** Comparison of hormone-based therapeutic strategies for melasma.

Therapeutic Category	Example Agents	Efficacy	Safety Considerations
SERMs	Raloxifene, Bazedoxifene	Significant improvement in pigmentation scores after 12 weeks in clinical trials.	Generally well-tolerated; long-term safety in melasma treatment requires further study.
Aromatase Inhibitors	-	Effective in reducing hyperpigmentation by decreasing estrogenic activity.	Risk of skin dryness, thinning, or atrophy with excessive estrogen suppression.
SPRMs	Asoprisnil (AP)	Suppresses extracellular melanin levels in vitro; reduces melanocyte activity.	Local application minimizes systemic exposure; long-term dermatological safety under investigation.
20-Hydroxyprogesterone Analogues	-	Potential to normalize pigmentation and reduce inflammation in melasma.	Topical formulations (e.g., nanocarriers) enhance safety and efficacy; minimal systemic absorption.

## Data Availability

No new data were created or analyzed in this study. Data sharing is not applicable to this article.

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
