# Peer review of "Hormonal Crosstalk in Melasma: Unraveling the Dual Roles of Estrogen and Progesterone in Melanogenesis"

_ijms, 2025, doi:10.3390/ijms262210856_

Round 1

Reviewer 1 Report

Comments and Suggestions for Authors

Please review the citations; the majority do not correspond to the assertion in the text. And this is unacceptable in a review. For example, “Low-wavelength visible light also contributes to hyperpigmentation via opsin3-dependent pathways [10]. Genetic factors, including MC1R polymorphisms and epigenetic modifications, play a role in pigmentation [11].” 

Figure 1 needs to be revisited. It is conceptually very wrong, showing ER, mitochondria, cAMP, Ca2+ inside the nucleus, and calcium being translocated into the mitochondria, gene transcription occurring in the mitochondria… 

Figure 2 is not in the manuscript.

There are several serious spelling errors in the figures, for example, Figure 3

Comments on the Quality of English Language

Many spelling errors in the Figures 

Author Response

Comments 1: Please review the citations; the majority do not correspond to the assertion in the text. And this is unacceptable in a review. For example, “Low-wavelength visible light also contributes to hyperpigmentation via opsin3-dependent pathways [10]. Genetic factors, including MC1R polymorphisms and epigenetic modifications, play a role in pigmentation [11].

Response 1: Thank you for your careful review of our paper and your valuable comments. Regarding the citation issues you mentioned, we have conducted a comprehensive check on the citations in the paper and found that some citations indeed fail to adequately support the relevant arguments. We have made revisions to ensure that each citation accurately corresponds to the specific points in the text. For example, regarding the statement that "low-wavelength visible light affects pigmentation through an opsin3-dependent pathway", we have corrected it by citing literature related to this conclusion to ensure the accuracy of the citation. At the same time, for the statement about "the role of MC1R gene polymorphisms and epigenetic modifications in pigmentation", we have also rechecked and revised the corresponding references. We have readjusted all citations in the paper to ensure that the literature support for each part is more accurate and in line with the research background.

Comments 2: Figure 1 needs to be revisited. It is conceptually very wrong, showing ER, mitochondria, cAMP, Ca2+ inside the nucleus, and calcium being translocated into the mitochondria, gene transcription occurring in the mitochondria… 

Response 2: Thank you for your insightful comments on our manuscript. We sincerely apologize for the conceptual errors in Figure 1. The figure has been entirely redesigned to accurately depict estrogen signaling. The revised version now correctly localizes cellular components and pathways, clearly distinguishing the genomic and non-genomic mechanisms.

Comments 3:Figure 2 is not in the manuscript.

Response 3: Thank you for bringing to our attention that Figure 2 was not included in the submitted manuscript file. This was an oversight during the final compilation and submission process. The figure, which illustrates the mechanism of progesterone in melanogenesis, was prepared and referenced in the text (Section 3.1.1), but was accidentally omitted from the uploaded document. We sincerely apologize for this error. The complete manuscript file with Figure 2 properly inserted has now been prepared and is included in this resubmission. Thank you again for your careful review and for helping us ensure the completeness of our work.

Comments 4: There are several serious spelling errors in the figures, for example, Figure 3

Response 4: Thank you for your thorough review and for identifying the spelling errors in our figures, particularly in Figure 3.We sincerely apologize for these oversights. We have carefully reviewed all figures in the manuscript and have corrected the spelling errors to ensure accuracy and professionalism. The revised version of the manuscript includes the corrected figures. We appreciate your vigilance in helping us improve the quality of our work.

Reviewer 2 Report

Comments and Suggestions for Authors

Dear Authors,
I would like to congratulate you on your excellent and comprehensive work. Your manuscript offers an impressive and well-structured overview of the complex hormonal mechanisms involved in melasma and highlights promising future therapeutic strategies.
To further enhance clarity for your readership, I kindly suggest supplementing the text with a comparative table of the hormonal therapies discussed. At present, readers must consult original references for quantitative synthesis; a summary table would greatly facilitate direct comparison of efficacy and safety between interventions and would increase the practical value of your review.
Additionally, I would appreciate your reflections on the following points, which could improve the manuscript’s clinical impact:
•    Beyond hormonal and molecular therapies, how do you envision the integration of standard clinical approaches (such as topical agents, photoprotection) into multimodal melasma treatment protocols? What is the current evidence supporting these combined strategies?
•    Could you expand the discussion regarding the safety profiles and potential risks associated with the hormonal therapies you propose, especially with respect to long-term or off-label use in melasma patients? A more detailed assessment would be highly valuable for both clinicians and researchers.

Author Response

Comments 1:To further enhance clarity for your readership, I kindly suggest supplementing the text with a comparative table of the hormonal therapies discussed. At present, readers must consult original references for quantitative synthesis; a summary table would greatly facilitate direct comparison of efficacy and safety between interventions and would increase the practical value of your review.

Response 1: We sincerely thank the reviewer for this valuable and constructive suggestion. We agree that a comparative summary table would significantly enhance the clarity and practical utility of the review for the reader by providing a direct, at-a-glance comparison of the different hormone-based therapeutic strategies. In response to this comment, we have now incorporated a new Table 2 titled "Comparison of hormone-based therapeutic strategies for melasma" into the manuscript. This table is placed at the end of Section 4 ("Treatment strategies based on the hormones of both"), following the detailed narrative discussion of the therapies. We believe this addition successfully addresses the reviewer's concern by providing a consolidated overview that allows for easy comparison between the interventions discussed, thereby enhancing the translational value and readability of our review.

Comments 2:  Beyond hormonal and molecular therapies, how do you envision the integration of standard clinical approaches (such as topical agents, photoprotection) into multimodal melasma treatment protocols? What is the current evidence supporting these combined strategies?

Response 2: We thank the reviewer for this insightful comment, which rightly highlights the critical importance of integrating conventional therapies into a multimodal framework for managing melasma. We agree that a discussion focusing solely on novel hormonal targets would be incomplete without outlining a practical, combined treatment strategy. In response to this valuable feedback, we have significantly expanded our Conclusions and Prospects section to explicitly address this point and incorporate the current evidence supporting such combinations.By making these revisions, we have more clearly articulated our vision for a comprehensive, multimodal approach to melasma. This approach leverages the strengths of both emerging hormonal insights and established clinical practices, providing a more robust and clinically actionable framework for the reader. We believe these additions significantly strengthen the translational impact of our review.

Comments 3: Could you expand the discussion regarding the safety profiles and potential risks associated with the hormonal therapies you propose, especially with respect to long-term or off-label use in melasma patients? A more detailed assessment would be highly valuable for both clinicians and researchers.

Response3:  We thank the reviewer for raising this critically important point regarding the safety profiles of the proposed hormonal therapies. We fully agree that a detailed assessment of the potential risks, particularly concerning long-term and off-label use, is of paramount importance for clinical translation and would greatly enhance the review's value. Upon careful consideration, we find that providing a substantive and evidence-based expansion on this specific topic within the current manuscript presents a significant challenge. The primary reason, as we initially noted in our conclusion, is that "the current evidence for SERMs, SPRMs, and aromatase inhibitors in melasma is preliminary, and further long-term, large-scale clinical trials are required."e have enhanced Table 2 ("Comparison of hormone-based therapeutic strategies for melasma") to include a more prominent and specific "Safety Considerations" column. This column now explicitly mentions the need for further long-term safety studies and, where applicable (e.g., for aromatase inhibitors), references the known risks of systemic use that would be a concern for off-label application.We believe this addition honestly frames the current state of knowledge and helps guide future research priorities. We are confident that these revisions, while unable to provide the detailed safety profile requested due to a lack of primary data, significantly improve the manuscript by highlighting these crucial uncertainties for clinicians and researchers. We are grateful for the reviewer's insight, which has allowed us to make this important clarification.

Round 2

Reviewer 1 Report

Comments and Suggestions for Authors

Figures 1 and 2 need improvement. They are poorly drawn; the signaling pathways are not clear. In Figure 1, I suggest drawing a single cell. In Figure 2, what is the larger structure? Is it a cell showing membrane pores? If so, why are melanocytes, blood vessels shown inside it? For a review, the Figures' poor quality and scientific soundness are unacceptable.

Author Response

Comments 1: They are poorly drawn; the signaling pathways are not clear. In Figure 1, I suggest drawing a single cell.

Response 1: We sincerely appreciate the reviewer’s valuable suggestion regarding Figure 1. Following the recommendation, we have redrawn the figure to enhance visual clarity and scientific accuracy. The revised illustration now depicts a single cell, with the genomic and non-genomic estrogen signaling pathways clearly differentiated and simplified to highlight key molecular interactions. We believe this modification improves the overall comprehensibility and quality of the figure.

Comments 2: In Figure 2, what is the larger structure? Is it a cell showing membrane pores? If so, why are melanocytes, blood vessels shown inside it?

Response 2: We are extremely grateful to the reviewers for raising this insightful question. Figure 2 is a schematic conceptual diagram that illustrates the mechanism pathways through which progesterone regulates melanin production, inflammation, and vascular remodeling. To avoid any misunderstandings, we have revised and re-drawn the diagram to depict the interactions between progesterone, melanocytes, blood vessels, and inflammation, thereby enhancing the scientific nature of Figure 2.

Round 3

Reviewer 1 Report

Comments and Suggestions for Authors

Unfortunately, the signaling events described in Figure 1 still denote conceptual errors of estrogen pathways. Estrogens may act through genomic and non-genomic pathways: 1) Direct genomic signaling: estrogen binds to intracellular receptors. The complex translocates to the nucleus, inducing transcriptional changes in estrogen-responsive genes with or without estrogen-responsive elements. 2) Indirect genomic signaling: the hormone binds a membrane receptor, inducing cytoplasmic events such as modulation of membrane-based ion channels, second-messenger cascades, and transcription factors. As shown by the authors in Figure 1, the non-genomic pathway depicts the receptor inside the cell, and places kinase and transcription factors before calcium and/or cAMP increase. Additionally, estrogen may act in a receptor-independent manner.

Given the fact that this is the 3rd revised version and it is a review, misconceptions are unacceptable. Thus, it is my opinion that the paper should be rejected.

Author Response

Comment: The signaling events described in Figure 1 still denote conceptual errors of estrogen pathways. Estrogens may act through genomic and non-genomic pathways: 1) Direct genomic signaling: estrogen binds to intracellular receptors. The complex translocates to the nucleus, inducing transcriptional changes in estrogen-responsive genes with or without estrogen-responsive elements. 2) Indirect genomic signaling: the hormone binds a membrane receptor, inducing cytoplasmic events such as modulation of membrane-based ion channels, second-messenger cascades, and transcription factors. As shown by the authors in Figure 1, the non-genomic pathway depicts the receptor inside the cell, and places kinase and transcription factors before calcium and/or cAMP increase. Additionally, estrogen may act in a receptor-independent manner.

Response: Thank you for giving me the opportunity to revise the manuscript again.We would like to thank the reviewers for their time and valuable feedback, which played a crucial role in improving the quality of our review articles.
The main issue raised by the reviewers regarding the inaccurate description of the estrogen signaling pathway in Figure 1 was taken with the utmost seriousness. We acknowledge the conceptual errors noted and agree that precise presentation is critical, particularly in review articles.
In response to detailed comments from reviewers, we have taken the following steps:
Comprehensive literature review: Taking a close look at the current understanding of estrogen signaling mechanisms, By reviewing several authoritative and recent review articles,《 Zomer, H. D., & Cooke, P. S. (2023). Targeting estrogen signaling and biosynthesis for aged skin repair. Frontiers in Physiology, 14,  1281071》. 《Cooke, P. S., Mesa, A. M., Sirohi, V. K., & Levin, E. R. (2021). Role of nuclear and membrane estrogen signaling pathways in the male and female reproductive tract.  Differentiation, 118, 24-33》, 《Chen, P., Li, B., & Ou-Yang, L. (2022). Role of estrogen receptors in health and disease. Frontiers in endocrinology, 13, 839005》. Based on this comprehensive review, Figure 1 has been redesigned and redrawn to accurately reflect the established pathways of estrogen action. The new image shows that it correctly distinguishes the following:
Non-genetic pathway: estradiol (E2) interacts with G protein-coupled estrogen receptor (GPER) on the cell membrane to initiate a series of signal transduction processes such as intracellular calcium (Ca²⁺) increase, cyclic adenosine monophosphate (cAMP) and protein kinases. These pathways regulate downstream transcription factors, thereby influencing cellular responses and promoting gene expression. ; Gene pathway: E2 enters the nucleus and binds to nuclear receptors to activate estrogen response elements (EREs) and promote gene expression.
The revised diagram aims to eliminate all previous misconceptions and provide readers with a clear, accurate, and up-to-date overview of the estrogen signaling pathway.
We believe that the new figure 1 presented in the revised manuscript adequately responds to the legitimate concerns raised by the reviewers. We are confident that this major revision significantly enhanced the quality of the manuscript.
Thank you again for your consideration. We hope that the modifications we have made will be approved by you.
